# Diffusion-Weighted MRI for Predicting Pathologic Complete Response in Neoadjuvant Immunotherapy

**DOI:** 10.3390/cancers14184436

**Published:** 2022-09-13

**Authors:** Wen Li, Nu N. Le, Natsuko Onishi, David C. Newitt, Lisa J. Wilmes, Jessica E. Gibbs, Julia Carmona-Bozo, Jiachao Liang, Savannah C. Partridge, Elissa R. Price, Bonnie N. Joe, John Kornak, Mark Jesus M. Magbanua, Rita Nanda, Barbara LeStage, Laura J. Esserman, Laura J. van’t Veer, Nola M. Hylton

**Affiliations:** 1Department of Radiology and Biomedical Imaging, University of California, 550 16th Street, San Francisco, CA 94158, USA; 2Department of Radiology, University of Washington, 1100 Fairview Ave N, Seattle, Washington, DC 98109, USA; 3Department of Epidemiology and Biostatistics, University of California, 550 16th Street, San Francisco, CA 94158, USA; 4Department of Laboratory Medicine, University of California, 2340 Sutter Street, San Francisco, CA 94115, USA; 5Department of Medicine, University of Chicago, Chicago, IL 60637, USA; 6I-SPY 2 Advocacy Group, 499 Illinois Street, San Francisco, CA 94158, USA; 7Department of Surgery, University of California, 550 16th Street, San Francisco, CA 94158, USA

**Keywords:** diffusion-weighted MRI, breast cancer, immunotherapy, pathologic complete response, neoadjuvant therapy

## Abstract

**Simple Summary:**

Immunotherapy targets patients’ immune systems to fight cancer. The aim of this retrospective study is to assess tumor response to pre-operative immunotherapy and predict pathologic complete response using MRI at an early treatment time- point. Based on our analysis with a cohort from the multi-center I-SPY 2 clinical trial, we found diffusion-weighted MRI is superior to dynamic contrast-enhanced MRI, where the latter is a standard and most-commonly used MRI modality, in assessing immunotherapy, while no significant difference was observed in the control cohort where only standard chemotherapy was provided.

**Abstract:**

This study tested the hypothesis that a change in the apparent diffusion coefficient (ADC) measured in diffusion-weighted MRI (DWI) is an independent imaging marker, and ADC performs better than functional tumor volume (FTV) for assessing treatment response in patients with locally advanced breast cancer receiving neoadjuvant immunotherapy. A total of 249 patients were randomized to standard neoadjuvant chemotherapy with pembrolizumab (pembro) or without pembrolizumab (control). DCE-MRI and DWI, performed prior to and 3 weeks after the start of treatment, were analyzed. Percent changes of tumor ADC metrics (mean, 5th to 95th percentiles of ADC histogram) and FTV were evaluated for the prediction of pathologic complete response (pCR) using a logistic regression model. The area under the ROC curve (AUC) estimated for the percent change in mean ADC was higher in the pembro cohort (0.73, 95% confidence interval [CI]: 0.52 to 0.93) than in the control cohort (0.63, 95% CI: 0.43 to 0.83). In the control cohort, the percent change of the 95th percentile ADC achieved the highest AUC, 0.69 (95% CI: 0.52 to 0.85). In the pembro cohort, the percent change of the 25th percentile ADC achieved the highest AUC, 0.75 (95% CI: 0.55 to 0.95). AUCs estimated for percent change of FTV were 0.61 (95% CI: 0.39 to 0.83) and 0.66 (95% CI: 0.47 to 0.85) for the pembro and control cohorts, respectively. Tumor ADC may perform better than FTV to predict pCR at an early treatment time-point during neoadjuvant immunotherapy.

## 1. Introduction

The number of clinical trials that investigate immunotherapeutic agents for breast cancer treatment has increased in recent years [1,2,3]. A notable result published from the I-SPY TRIAL (Investigation of Serial studies to Predict Your Therapeutic Response with Imaging And moLecular Analysis) showed that the addition of pembrolizumab to standard neoadjuvant chemotherapy (NAC) achieved a higher estimated pathologic complete response (pCR) rate than standard NAC alone [3]. Recently, the FDA approved pembrolizumab in combination with chemotherapy for high-risk early-stage triple-negative breast cancer [4].

Immunotherapy works by boosting the patient’s own immune system to fight cancer cells, a mechanism of action which differs from that of conventional NAC. Pembrolizumab is a PD-1 (programmed death-1) inhibitor that triggers the activation of the T-cell immune response against cancer cells [5]. In the neoadjuvant setting, the reported pCR rates for immunotherapy (alone or in combination with standard NAC) range from 43.5% to 64.8% [6], which means approximately half of the patients treated with immunotherapy do not achieve pCR. Studies showed that the patients who achieve pCR after NAC had better recurrence-free survival than patients who do not achieve pCR [7,8,9]. Since not all of the patients benefit from immunotherapy and some patients may experience serious side effects [6,10], it is crucial to assess tumor response early during treatment. However, assessing the response to immunotherapy using conventional imaging methods, such as measuring changes in tumor size according to the Response Evaluation Criteria in Solid Tumors (RECIST), is limited because RECIST was designed to assess the response to standard therapies [11,12]. Pseudo-progression with immunotherapy is often observed, and is characterized by an initial increase in tumor size followed by a decrease in tumor size or volume [13,14,15].

Diffusion-weighted MRI (DWI) is a non-invasive, non-contrast MRI technique that is different from dynamic contrast-enhanced (DCE) MRI. The apparent diffusion coefficient (ADC) calculated from DWI, reflects the water diffusion within the tumor microenvironment and the ADC values are associated with the cellularity and integrity of cell membranes. Results from the multi-center ACRIN 6698 clinical trial showed that the percent change of breast tumor ADC measured 12 weeks after the start of NAC is predictive of pCR [16]. A subsequent multi-center study showed that ADC adds independent value to functional tumor volume (FTV) measured from DCE-MRI in the prediction of pCR [17]. FTV is a well-established imaging marker of treatment response demonstrated by the multi-center I-SPY 1/ACRIN 6657 clinical trial [18,19]. In I-SPY 2, FTV continues to be used as an integral imaging marker for monitoring treatment response. Each patient who enrolls in the I-SPY 2 trial undergoes MRI scans including both DCE and DWI at four treatment time-points. However, the optimum use of DCE and/or DWI in assessing response to immunotherapy is still unclear. In this study, we compared the predictive performance of ADC with that of FTV, both measured at an early treatment time-point, for patients treated with pembrolizumab in addition to standard chemotherapy and for patients treated with standard chemotherapy alone in I-SPY 2. As a secondary aim, we explored the histogram percentiles as an alternative ADC metric other than mean ADC to address tumor heterogeneity as a potential imaging marker to predict pCR.

## 2. Materials and Methods

### 2.1. Patient Population

The patient population of this study was a sub-cohort from the I-SPY 2 trial, an adaptively randomized phase 2 neoadjuvant clinical trial for early-stage high-risk breast cancer (ClinicalTrials.gov ID: NCT01042379) [20]. I-SPY 2 is a multi-site clinical trial and each participating institute obtained Institutional Review Board (IRB) approval independently. Written consents for both the screening and treatment were required for each patient. Patients with HER2-negative breast cancer who enrolled in I-SPY 2 consecutively between 26 November 2015 and 5 November 2016 were eligible to be randomized to paclitaxel + pembrolizumab for 12 weeks followed by four cycles of doxorubicin plus 600 mg/m^2^ intravenous cyclophosphamide (AC) every 2 to 3 weeks (*n* = 69, referred to as the “pembro” arm below). The control group consisted of 180 patients who had HER2-negative breast cancer and were randomized to receive standard chemotherapy (paclitaxel followed by AC). The sample sizes of the control and pembro groups were determined by the I-SPY 2 drug graduation rule. Details of the study design related to this patient population in I-SPY 2 were published previously [21].

### 2.2. MRI Analysis

Breast MRIs were performed at participating institutions on whole body MRI scanners from different vendors, at field strengths of 1.5 T or 3.0 T using a dedicated breast coil. Studies at baseline/prior to treatment (T0) and 3 weeks after start of treatment (T1) were used for this analysis of early prediction of treatment response. Additional MRI exams performed at inter-regimen (approximately 12 weeks after the therapy started) and before surgery (approximately 24 weeks after the therapy started) were not included in this analysis. The same scanner, MRI breast coil, and sequences were used to scan the same patient across the multiple time-points. Details of the I-SPY 2 MRI protocol were previously published [17]. Briefly, the I-SPY 2 MRI scan protocol used for this cohort included bilateral axial acquisition of a T2-weighted sequence, a pre-contrast echo planar imaging (EPI) DWI sequence, and DCE using a bilateral, 3D, fat-suppressed, and T1-weighted sequence with 80–100 s temporal resolution. DWI was acquired using two b-values: 0 and 800 s/mm^2^ with an acquisition time ≤ 5 min.

ADC maps were calculated from DWI using in-house software developed in IDL (Exelis Visual Information Solutions, Boulder, CO, USA) based on the monoexponential decay model: $ADC=-\ln\frac{S_{b}}{S_{0}}/b$, where b = 800 s/mm^2^ and S_0_ is the signal intensity corresponding to no diffusion gradients (b = 0 s/mm^2^). The DWI data and corresponding ADC maps were reviewed, and the tumor region-of-interest (ROI) was manually drawn by two readers with 7 (W.L.) and 0.2 (N.L.) years of experience in breast DWI analysis. Both readers were blinded to the pathologic outcomes. Each reader was trained on a separate cohort of 30 patients with DWI acquired at baseline and 3-week time-points. Discrepancies between the readers were resolved after consultation of a breast radiologist with 10 years of experience (N.O.). After the training, both of the readers assessed all of the DWI in this study independently and the corresponding report of the reader study was published [22]. The ROIs in the present study were multiple-slice restricted ROIs, defined as the most diffusion-restricted area on DWI (dark on ADC map and bright on b = 800 s/mm^2^ diffusion-weighted image) in all of the axial slices in which the tumor can be identified [22]. Based on the agreement found in the reader study, multiple-slice restricted ROIs from one reader (W.L.) were used in this study. To avoid including volumes with T2 shine-through effect, the ROI placement was strictly limited to the volume with high signal intensity on b = 800 s/mm^2^ diffusion-weighed image with low ADC values. In addition, special care was taken to avoid cystic, necrotic or fatty components by excluding areas with no contrast enhancement on DCE-MRI [23]. The ROIs at 3-week were placed at similar locations to the baseline MRI, if possible, given the differences in breast position and treatment response, to be able to reflect the treatment response. ADC metrics were quantified using estimated mean and percentiles (minimum, 5th, 15th, 25th, 50th, 75th, 95th, maximum) from ADC histograms extracted from ROIs. In addition, DWIs were ranked based on image quality in both b = 0 and b = 800 images based on fat suppression, artifacts, and signal-to-noise ratio. An overall quality rating of poor, moderate, or good was assigned to each DWI and the poor quality DWIs were excluded from the analysis.

FTV (in cubic centimeter, cc) was also calculated using DCE-MRI as a standard imaging marker in I-SPY 2. Specifically, percent enhancement (PE, in %) and signal enhancement ratio (SER) maps were derived from DCE-MRI using T1-weighted images acquired before and multiple times after the injection of contrast agent. PE was calculated as the percent change of signal intensity between pre-contrast and early (nominal 2.5 min after injection) post-contrast: $PE=\frac{S_{1}-S_{0}}{S_{0}}\times 100$. SER was calculated as the ratio of early enhancement to late (nominal 7.5 min after) enhancement: $SER=\frac{S_{1}-S_{0}}{S_{2}-S_{0}}$, where $S_{0},\;S_{1},\;\mathrm{and}\;S_{2}$ represent the signal intensities of each voxel in the pre-contrast, early post-contrast, and late post-contrast images, respectively. FTV was computed as the combined volume of all voxels with PE ≥ 70% and SER ≥ 0.

### 2.3. Treatment Outcome

After completion of NAC, patients underwent surgery (lumpectomy or mastectomy). The surgical specimens were analyzed for response by site pathologists trained to assess residual invasive disease, with pCR defined as no invasive residual disease in primary breast and lymph nodes. The binary pathological outcome (pCR versus non-pCR) was used as the treatment outcome in this study. The patients who exited the trial without surgery or did not undergo surgery for various reasons were excluded from our analysis.

### 2.4. Statistical Analysis

Multiple MRI tumor metrics, calculated from DWI and DCE-MRI, were evaluated for the prediction of pCR. These metrics were: percent change of ADC metrics (mean, minimum, maximum, as well as 5th, 15th, 25th, 50th, 75th, 95th percentile) and percent change of FTV between T0 and T1 were considered as potential MRI variables (herein referred to as ADC_mean, ADC_min, ADC_max, ADC_5, ADC_15, ADC_25, ADC_50, ADC_75, ADC_95, FTV, respectively) for the prediction of pCR. To enable a direct comparison between ADC and FTV, the percent change of ADC was calculated as: [(ADC at T1)–(ADC at T0)]/(ADC at T0). The percent change of FTV was calculated as: [(FTV at T1)–(FTV at T0)]/(FTV at T0). According to our observation, the distribution of percent change of ADC and percent change of FTV were skewed so their values were given as a median with the interquartile range. The associations between MRI variables and pCR were analyzed by single predictor or multiple predictors in a logistic regression model. The area under the receiver operating characteristic curve (AUC) was used to assess the predictive performance of a single-predictor model. Hypothesis tests of the differences between two AUCs were evaluated using the DeLong test [24]. The *p*-values of the differences between MRI variables in pCR versus non-pCR groups were estimated using the Wilcoxon rank sum test. The impact of FTV and treatment on the association between ADC with pCR was studied using nested models where FTV and treatment were added one-at-a-time as covariate to the model starting with ADC alone. The odds ratio of percent change of ADC or FTV for pCR was calculated per 10% increments in the logistic regression model. Correlation coefficients and their significant levels (*p*-values) between two MRI variables were tested by Spearman’s rank correlation test. To compare the predictive performance of MRI variables, the full analysis cohort was stratified by treatment arm: control (patients received paclitaxel only as the first regimen) and pembro (patients received paclitaxel + pembrolizumab as the first regimen). Statistical analysis was implemented in R version 4.1.3 and the statistical significance level was set at α = 0.05.

## 3. Results

A total of 249 patients were considered for inclusion in the analysis. One hundred and eighty patients were randomized to be the controls and 69 patients to the pembro. Figure 1 shows a flowchart of data exclusion and inclusion. In summary, 103 total patients were included in the analysis (75 in controls and 28 in pembro) and the majority of the exclusions (117 patients in total) are due to the poor quality of DWI at either or both T0 and T1. The patient characteristics of the excluded and included cohorts are shown in Table 1. The included patients were on average 3 years younger than those who were excluded (*p* = 0.004). Race, HR/HER2 subtype, treatment, or menopausal status between the included and excluded cohorts were balanced between the two groups.

Percent changes of ADC metrics at T1 compared to T0 are shown in Table 2. The values of ADC metrics and FTV at T0 and T1 can be found in the Appendix A, Appendix A. In general, the percent changes in tumor ADC were positive, which indicates that ADC increased after 3 weeks of treatment. In the full cohort of 103 patients, statistically significant differences between the pCR and non-pCR groups were observed in ADC_mean, ADC_min, and lower percentiles of ADC histogram—up to the 50th percentile. In the control cohort, a statistically significant difference was only observed at ADC_95. In the pembro cohort, statistically significant differences were observed at ADC_mean, ADC_min, ADC_15, and ADC_25. Percent changes of FTV in the full cohort and in individual treatment cohorts are listed in Table 2. FTV decreased in all of the cohorts despite pCR outcomes. The difference between pCR and non-pCR groups reached statistically significance (*p* = 0.016) in the full cohort but not in the sub-cohorts.

Figure 2 shows boxplots of the percent change of ADC and FTV at T1 by pCR outcome and cohort. Although both the ADC and FTV indicated a good separation between the pCR and non-pCR groups in the full cohort, ADC was visually better separated than FTV in the pembro cohort with *p*-values of 0.041 and 0.34 for ADC and FTV, respectively (Table 2). The boxplots suggested stronger differences were found for ADC_mean between the pCR and non-pCR groups.

The AUCs for the percent change of the ADC metrics evaluated to predict pCR are listed in Table 3. In the full cohort, the AUCs ranged from 0.49 to 0.67, with the highest AUCs for ADC_mean and ADC_min. When the full cohort was stratified by treatment, the AUCs ranged from 0.55 to 0.69 and 0.47 to 0.75 for the control and pembro cohorts, respectively. The highest AUCs were observed for ADC_95 and ADC_25 percentiles in the control and pembro cohorts, respectively. Although the highest AUCs were not observed at ADC_mean, the difference between the ADC_mean and ADC metrics that yielded the highest AUCs did not reach the pre-defined statistical significance level either in the control (AUC = 0.63 versus AUC = 0.69, *p* = 0.43) cohort or in the pembro (AUC = 0.73 versus AUC = 0.75, *p* = 0.62) cohort.

The AUCs for the percent change of FTV were estimated as 0.65 (95% CI: 0.52 to 0.78), 0.66 (95% CI: 0.47 to 0.85), and 0.61 (95% CI: 0.39 to 0.83) in the full cohort, control, and pembro, respectively (Table 3). Figure 3 displays the ROC curves for ADC and FTV. The biggest separation of two ROC curves was observed in the pembro cohort, suggesting a higher performance for ADC in this cohort.

Table 4 shows the odds ratios, 95% CI thereof, and the *p*-values for ADC_mean in the logistic regression model of predicting pCR, where ADC_mean was a single predictor (left column), FTV was added (middle column), and FTV and treatment were both added to the model (right column). The odds ratios were estimated in the full cohort (*n =* 103, pCR rate = 29%). Table 4 shows that ADC_mean had a statistically significant association with pCR even in the multivariate models. For comparison, the effect of ADC_mean and treatment to FTV is shown in the last row of Table 4, where the significance of FTV changed (*p*-value from 0.039 to 0.045) after ADC_mean was added to the model.

The correlations between ADC_mean and FTV were weak and not statistically significant.: *r* = 0.041 (*p* = 0.68) in the full cohort, *r* = −0.089 (*p* = 0.45) in control, and *r* = 0.29 (*p* = 0.13) in pembro.

Figure 4 and Figure 5 show example cases of pCR and non-pCR. In the pCR case, the tumor FTV was 163.9cc at T0 and 155.6cc at T1 (5% decrease). ADC_mean was 1.102 × 10^−3^ mm^2^/s at T0 and 1.437 × 10^−3^ mm^2^/s at T1 (30% increase). In the non-pCR case, the tumor FTV was 3.5cc at T0 and 0.6cc at T1 (82% decrease), while mean ADC was 0.819 × 10^−3^ mm^2^/s and 0.831 × 10^−3^ mm^2^/s, respectively (1% increase). ADC changed more than FTV in the pCR case and less than FTV in the non-pCR case. Additional example cases from the control cohort are provided in the Appendix A, Appendix A.

## 4. Discussion

This study measured the percent change of ADC metrics (ADC_mean, ADC_min, ADC_5, ADC_15, ADC_25, ADC_50, ADC_75, ADC_95, maximum) before and after 3 weeks of neoadjuvant therapy in breast cancer patients enrolled in the I-SPY 2 trial. We assessed the prediction of pCR using ADC metrics measured in patients treated with standard NAC alone or in combination with immunotherapy. We also compared the predictive performance of mean ADC with FTV measured from DCE-MRI. We found that all of the ADC metrics except ADC_95 and ADC_max performed better than FTV in the immunotherapy cohort after 3 weeks of therapy.

This study found that the change in the ADC_mean was predictive of pCR. This is consistent with our prior work, in which the ADC_mean was also predictive of pCR at T1 [17]. The previous study also included data from a larger cohort of patients in the I-SPY 2 TRIAL, (*n =* 328 at T1) but there was no overlap between that cohort and the cohort used in this analysis. The AUC reported previously was 0.57, while it was 0.67 in this analysis. Using a different ROI delineation strategy may have contributed to improved AUC in this study. The use of a single drug arm with controls in this analysis may also have contributed to the improved AUC, since the previous study included patients from four different experimental drug arms in I-SPY 2, with no controls.

In addition to the ADC_mean, we found that changes of lower percentiles (ADC_min, ADC_5, ADC_15, ADC_25, ADC_50) were statistically significant and estimated AUCs were higher for the lower percentiles compared to the higher percentiles. This finding aligns with results reported previously, where lower percentiles seemed more predictive than higher percentiles for treatment response [25,26]. This could be explained by the hypothesis that low ADC may reflect tumor regions of higher cell density that are more sensitive to chemotherapy [27,28,29,30]. However, we did not observe this trend in the control cohort. This may indicate response heterogeneity within the tumor.

In the pembro cohort, FTV performed poorly and the majority of the ADC metrics achieved a higher AUC than FTV. However, ADC did not achieve a higher AUC in the control cohort. This finding may indicate that FTV may be limited in assessing the tumor response to immunotherapy at the early treatment time-point. Previous studies reported that size measurements may not accurately assess the tumor response to immunotherapy due to possible pseudo-progression at early treatment time-points [13,31,32]. Because pseudo-progression has been histologically proved to be infiltration and recruitment of various immune cells including lymphocytes in the tumor, the change of tumor ADC at 3 weeks might reflect the change in cellularity due to the immune cell infiltration, especially the change in lower ADC percentiles (i.e., highly cellular portion).

To our knowledge, this is the first multi-center study to assess tumor ADC in the prediction of treatment response to immunotherapy compared to standard NAC in breast cancer. The results of our study suggest that DWI can be used as a complementary method to DCE-MRI in assessing the response to NAC. However, this study has several limitations. First, the sample size is small, especially in the pembro cohort. The number of patients included in the analysis was 28 (15 pCRs; 13 non-pCRs), which prohibited further analysis by breast cancer subtype. Second, a large number of DWI exams were excluded due to poor image quality, further limiting the sample size for analysis. A total of 117 patients (47%) were excluded because of poor image quality at T0, T1, or both. Most of the excluded cases (*n =* 102; 32 pembro and 70 controls) had poor-quality DWI in both visits. This high exclusion rate clearly calls for a standardized quality control for DWI. Third, the patients randomized to pembrolizumab were treated by standard NAC plus pembrolizumab so the comparison between the pembro and control cohorts was not purely immunotherapy versus standard NAC. Finally, manual delineation of ROIs and qualitative ranking of DWI quality were subjective and user-dependent. An intelligent and objective quality ranking method for multi-center DWI studies is needed.

## 5. Conclusions

In conclusion, our findings suggest that at an early treatment time-point, mean ADC performs better than FTV in the prediction of pCR for patients who received neoadjuvant immunotherapy. In addition, we also found that ADC histogram percentiles may achieve a better predictive performance than mean ADC if they were selected based on treatment. Future studies are needed to determine the optimal way of incorporating ADC into the clinical workflow in combination with FTV or replacing FTV as an alternative imaging marker to maximize the prediction of treatment response using MRI in patients receiving immunotherapy.

## Figures and Tables

**Figure 1 cancers-14-04436-f001:**
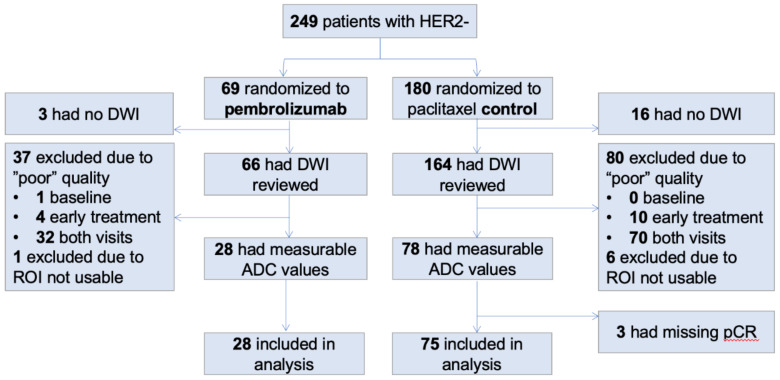
Data inclusion and exclusion. Out of 249 patients who had HER2- breast cancer, 103 patients were included in our analysis.

**Figure 2 cancers-14-04436-f002:**
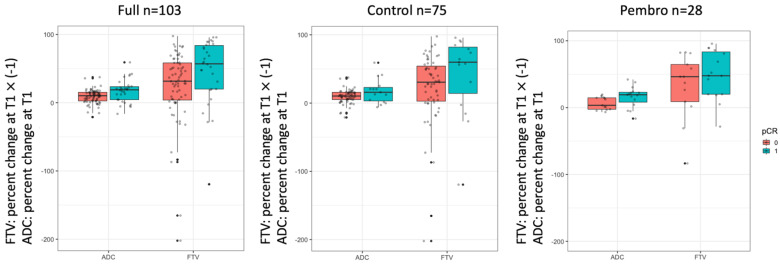
Boxplots of percent change in mean ADC and FTV at 3-week. The percent change in FTV was multiplied by −1 so a positive change in both ADC and FTV means positive response to treatment. Each dot represents an individual patient. ADC: apparent diffusion coefficient. FTV: functional tumor volume. pCR = 0 are non–pCRs and pCR = 1 are pCRs.

**Figure 3 cancers-14-04436-f003:**
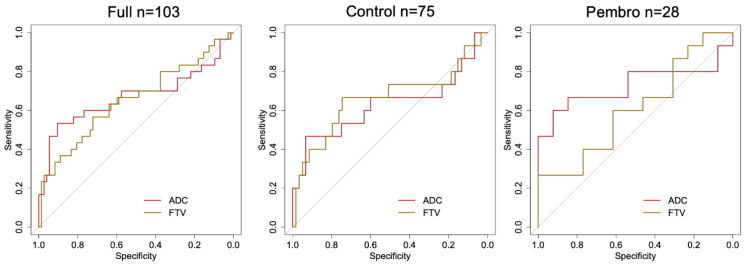
ROC curves of percent change in mean ADC and FTV for predicting pCR at 3-week. In the full cohort (left), pCR rate is 29% (*n =* 30) and the area under the ROC curve is 0.67 for ADC and 0.65 for FTV. In the control cohort (middle), pCR rate is 20% (*n =* 15) and the area under the ROC curve is 0.63 for ADC and 0.66 for FTV. In the pembro cohort (right), pCR rate is 54% (*n =* 15) and the area under the ROC curve is 0.73 for ADC and 0.61 for FTV. ADC: apparent diffusion coefficient. FTV: functional tumor volume. pCR: pathologic complete response.

**Figure 4 cancers-14-04436-f004:**
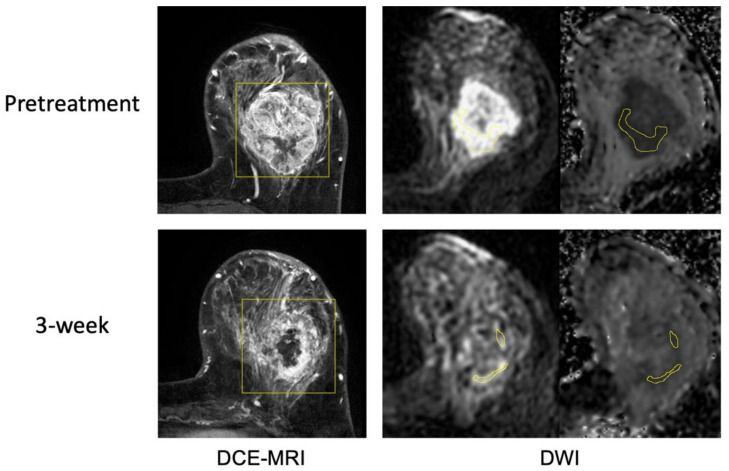
Example case 1 with a pathologic complete response. The patient was treated by paclitaxel + pembrolizumab followed by cyclophosphamide. Representative MR images are shown from pretreatment (top row) and at 3-week (bottom row). Images from DCE-MRI were acquired 144 s after the contrast injection. Images from DWI are shown in a pair of original DWI (b = 800 s/mm^2^) and ADC map. ROIs are shown in yellow (rectangular box in DCE and hand-drawn in DWI).

**Figure 5 cancers-14-04436-f005:**
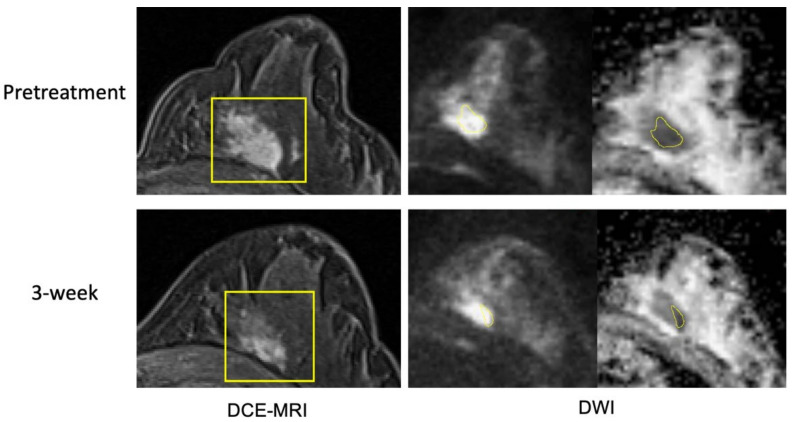
Example case 2 without a pathologic complete response. The patient was treated by paclitaxel + pembrolizumab followed by cyclophosphamide. Representative MR images are shown from pretreatment (top row) and at 3-week (bottom row). Images from DCE-MRI were acquired 119 s after the contrast injection. Images from DWI are shown in a pair of original DWI (b = 800 s/mm^2^) and ADC map. ROIs are shown in yellow (rectangular box in DCE and hand-drawn in DWI). Images from pretreatment and at 3-week may seem different by visual comparison due to different positioning in the MRI scanner. Breast tissue in DCE-MRI suggests that they are from similar slice location.

**Table 1 cancers-14-04436-t001:** Patient characteristics.

	Included (*N* = 103)	Excluded (*N* = 146)	*p* *
Age	46 +/− 11	49 +/− 11	0.004
Race			0.54
White	76 (74%)	115 (79%)	
Black or African American	15 (15%)	17 (12%)	
Asian	8 (8%)	8 (5%)	
American Indian or Alaska Native	0 (0%)	3 (2%)	
Others	3 (3%)	3 (2%)	
Subtype			0.12
HR+	62 (60%)	72 (49%)	
HR-	41 (40%)	73 (50%)	
Treatment			0.89
Paclitaxel	75 (73%)	105 (72%)	
Paclitaxel + Pembrolizumab	28 (27%)	41 (28%)	
Menopausal Status			0.26
Premenopausal	58 (56%)	68 (47%)	
Postmenopausal	26 (25%)	49 (34%)	
Perimenopausal	7 (7%)	6 (4%)	
Others	12 (12%)	23 (16%)	

* Assessed by Student’s *t*-test for age and Fisher’s test for categorical variables.

**Table 2 cancers-14-04436-t002:** Percent change MRI variables after 3 weeks of therapy.

MRI Variable	Full (*n =* 103)			Control (*n =* 75)			Pembro (*n =* 28)		
	pCR (*n =* 30)	Non-pCR (*n =* 73)	*p*	pCR (*n =* 15)	Non-pCR (*n =* 60)	*p*	pCR (*n =* 15)	Non-pCR (*n =* 13)	*p*
ADC_mean	18.9 (4.4, 23.6)	10.2 (2.6, 15.3)	0.0076	15.8 (3.1, 22.9)	10.3 (5.1, 15.8)	0.13	19.2 (8.2, 23.1)	3.5 (−2.4, 14.7)	0.041
ADC_min	28.0 (5.9, 54.7)	12.6 (−0.7, 24.6)	0.0075	23.4 (−2.7, 53.0)	14.1 (−0.8, 19.4)	0.15	29.7 (10.6, 56.5)	9.6 (−0.2, 20.8)	0.029
ADC_5	20.0 (−1.1, 32.8)	8.0 (0, 17.5)	0.0297	18.2 (−6.5, 33.5)	9.1 (−0.3, 19.4)	0.35	23.5 (4.7, 32.1)	6.1 (4.6, 10.5)	0.0799
ADC_15	17.3 (0.3, 27.6)	9.6 (2.4, 16.0)	0.039	14.6 (0.6, 26.7)	9.9 (3.3, 17.2)	0.47	22.2 (2.2, 28.3)	7.8 (1.1, 11.2)	0.040
ADC_25	17.9 (3.1, 29.7)	8.6 (2.4, 15.4)	0.014	15.7 (1.8, 23.2)	10.2 (3.6, 16.2)	0.25	23.6 (5.3, 31.4)	5.2 (0, 11.9)	0.025
ADC_50	18.0 (3.0, 23.5)	10.1 (3.2, 15.5)	0.048	13.2 (0.2, 23.8)	10.4 (3.8, 15.5)	0.34	19.5 (7.1, 22.996)	2.9 (−1.2, 14.2)	0.098
ADC_75	14.9 (1.5, 23.7)	8.5 (1.8, 15.6)	0.052	14.6 (1.1, 25.3)	8.9 (2.8, 14.9)	0.19	15.2 (5.5, 21.1)	1.8 (−3.6, 17.4)	0.196
ADC_95	15.0 (1.1, 20.4)	7.3 (0, 14.0)	0.087	18.6 (7.9, 22.6)	7.5 (1.5, 14.3)	0.027	6.01 (0.5, 16.1)	0.8 (−4.98, 12.5)	0.39
ADC_max	6.3 (−1.2, 11.8)	7.4 (−1.7, 13.8)	0.85	10.8 (1.8, 14.8)	7.5 (−2.2, 14.6)	0.53	3.8 (−5.5, 9.7)	3.8 (0.7, 9.9)	0.82
FTV	−56.97 (−83.9, −19.98)	−31.5 (−58.4, −3.8)	**0.016**	−59.9 (−82.0, −14.2,)	−30.6 (−54.2, −2.7)	0.056	−47.7 (−83.4, −20)	−46.3 (−64.6, −8.99)	0.34

**Table 3 cancers-14-04436-t003:** Area under the ROC curve for percent change of MRI variables at 3-week.

MRI Variable	Full (*n =* 103, pCR Rate 29%)	Control (*n =* 75, pCR Rate 20%)	Pembro (*n =* 28, pCR Rate 54%)
ADC_mean	0.67 (0.53, 0.81)	0.63 (0.43, 0.83)	0.73 (0.52, 0.93)
ADC_min	0.67 (0.53, 0.80)	0.62 (0.42, 0.82)	0.74 (0.55, 0.93)
ADC_5	0.64 (0.496, 0.78)	0.58 (0.38, 0.78)	0.697 (0.48, 0.91)
ADC_15	0.63 (0.49, 0.77)	0.56 (0.36, 0.76)	0.73 (0.52, 0.94)
ADC_25	0.66 (0.51, 0.80)	0.597 (0.397, 0.797)	0.75 (0.55, 0.95)
ADC_50	0.62 (0.49, 0.76)	0.58 (0.38, 0.78)	0.69 (0.48, 0.897)
ADC_75	0.62 (0.49, 0.76)	0.61 (0.41, 0.81)	0.65 (0.43, 0.86)
ADC_95	0.61 (0.48, 0.74)	0.69 (0.52, 0.85)	0.60 (0.38, 0.82)
ADC_max	0.49 (0.37, 0.61)	0.55 (0.39, 0.71)	0.47 (0.24, 0.70)
FTV	0.65 (0.52, 0.78)	0.66 (0.47, 0.85)	0.61 (0.39, 0.83)

**Table 4 cancers-14-04436-t004:** Odds ratio of percent change in mean ADC in logistic regression model (outcome: pCR).

MRI Variable	Univariate		Multivariate: ADC + FTV		Multivariate: ADC + FTV + Treatment	
	Odds Ratio	*p*	Odds Ratio	*p*	Odds Ratio	*p*
Mean ADC	1.67 (1.17, 2.5)	0.0039	1.70 (1.18, 2.59)	0.0032	1.71 (1.18, 2.62)	0.0032
FTV	0.90 (0.80, 0.995)	0.039	0.90 (0.80, 0.998)	0.045	0.92 (0.81, 1.02)	0.045

## Data Availability

Imaging and clinical data from this study will be made available upon request in accordance with I-SPY 2 Data and Publication Committee policies.

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
