# Peer review of "Diffusion-Weighted MRI for Predicting Pathologic Complete Response in Neoadjuvant Immunotherapy"

_cancers, 2022, doi:10.3390/cancers14184436_

Round 1

Reviewer 1 Report

The authors proposed the value of ADC in accessing the immunotherapy treatment response. The potential value of this study is appreciated. However, several issues need to be addressed:

1.     The manuscript is overly wordy and difficult to read. It requires thorough editing and polishing with a focus on sentence structure and formatting.

2.     The rationale of this study needs to be clarified. I did not see the importance of the 5th to 95th percentile of ADC values.

Author Response

We would like to thank Reviewer 1’s feedback and comments. Please see our response below.

  1. The manuscript is overly wordy and difficult to read. It requires thorough editing and polishing with a focus on sentence structure and formatting.

Response: We appreciate the Reviewer’s feedback. We did substantial editing to the original manuscript. Hopefully the revised version is easier to read.

  1. The rationale of this study needs to be clarified. I did not see the importance of the 5thto 95th percentile of ADC values.

Response: We would like to thank the Reviewer’s comments. We think histogram percentiles may provide additional information for tumor heterogeneity so we added a sentence at the end of Introduction to point out that it was included as a secondary aim of this study.

Reviewer 2 Report

In this submission to Cancer, the authors conclude that DWI MRI can be a predicting pCR in Neoadjuvant Immunotherapy. The authors results reveal that ADC_min and ADC_25 in Pembro group has a highest AUC as compared to the Control and instead of FTV, DWI can be better predictive marker although the AUC is not very high.

I consider this manuscript to be of interest to readers of this journal. As such, I support the publication of the manuscript with a minor but required revision.

1. Line 21 Simple Summary: why this section is missing

2. In abstract line 29 should be rewritten as FTV at baseline and at 3 weeks were evaluated….

Introduction:

Line 64 is a repetition.

Line 65 more reference should be cited.

Line 67, DWI is not a functional MRI technique. Rewrite the sentence.

DWI abbreviation is repeated twice in line 80

Line 77, from Each patient…. Till line 80 are suited for the Methods section than introduction; not pertinent to why the study is being conducted

Materials and Methods:

1.     How was the sample size estimated?

            MRI analysis:

1.     How many patients were having Cyst, necrosis, and fatty component?

2.     How were the ROIs placed at the similar locations to the baseline MRI?

Statistical Analysis:

1.     Why not the authors use this [(FTV at T1) – (FTV at T0)] / (FTV at T0) instead of  [(FTV at T0) – (FTV at T1)] / (FTV at T0) to calculate the percent change in FTV? Make it consistent with ADC percent change calculation.

Results:

1.     I recommend the authors to discuss Table 2 first and then Table 3. It’s confusing to read.

2.     Line 219: Is this standard? A decrease in FTV should be reported as a negative.

3.     From Supplemental Tables S1 and S2, it appears that % change FTV is a much better differentiator between pCR and non-pCR than in shown in Table 2. While Table 2 reports a similar median change in FTV between the two groups (47.7 vs. 46.3), tables S1 and S2 show that the change in median FTV is very different between the two groups for the pembro cohort (70.5% vs. 49.2%). Why is that?

4.     Similarly, the median ADC values in Table S2 for non-pCR and pCR groups are very similar at T0 and T1, how could the percent change has a low p-value for the change in ADC_mean in Pembro group between pCR and Non-pCR

5.     What is “FTV adjusted for treatment” in Table 4. I don’t understand how Table 4 was made or its subsequent analysis; what do the odds ratios represent? What are the differences between the columns?

6.     A color bar should be added to Figure 4 and Figure 5

7.     Why were the ROIs drawn on DCE-MRI were not copied to ADC maps why drawing manual ROIs were considered.

8.     Authors mentioned in the MRI analysis section line 134 that ROIs were placed at the similar locations to the baseline MRI. Figure 4 doesn’t depict the same.

9.     As per my visual inspection the slice in the Figure 5 pretreatment and 3 weeks seems unmatched. 

10.  Secondly Figure 5 is an example without a pathologic complete response but why the enhancement is less at 3 weeks as compared to pretreatment. Can authors present a better case example?

11.  I recommend authors to present case examples for control pCR and non-pCR

Discussion:

1.      I recommend authors to rewrite the discussion section with just discussing the results instead of explaining them. Example: Para 6th and 7th seems no relevance with the discussion section.

Round 2

Reviewer 1 Report

Several formatting issues were discovered in the downloaded version of this article, such as the duplication of tables 2 and 4. I am otherwise pleased to accept it.